# Pancreatic Cancer Associated Fibroblasts (CAF): Under-Explored Target for Pancreatic Cancer Treatment

**DOI:** 10.3390/cancers12051347

**Published:** 2020-05-25

**Authors:** Jeffrey Norton, Deshka Foster, Malini Chinta, Ashley Titan, Michael Longaker

**Affiliations:** 1Hagey Laboratory for Pediatric Regenerative Medicine, Division of Plastic and Reconstructive Surgery, Stanford University School of Medicine, Stanford, CA 94305, USA; dsfoster@stanford.edu (D.F.); mchinta@stanford.edu (M.C.); atitan@stanford.edu (A.T.); longaker@stanford.edu (M.L.); 2Division of General Surgery, Department of Surgery, Stanford University School of Medicine, Stanford, CA 94305, USA; 3Institute for Stem Cell Biology and Regenerative Medicine, Stanford University School of Medicine, Stanford, CA 94305, USA

**Keywords:** cancer associated fibroblast (CAF), pancreatic ductal adenocarcinoma (PDAC)

## Abstract

Pancreatic cancer is the 4th leading cause of cancer deaths in the United States. The pancreatic cancer phenotype is primarily a consequence of oncogenes disturbing the resident pancreas parenchymal cell repair program. Many solid tumor types including pancreatic cancer have severe tumor fibrosis called desmoplasia. Desmoplastic stroma is coopted by the tumor as a support structure and CAFs aid in tumor growth, invasion, and metastases. This stroma is caused by cancer associated fibroblasts (CAFs), which lay down extensive connective tissue in and around the tumor cells. CAFs represent a heterogeneous population of cells that produce various paracrine molecules such as transforming growth factor-beta (TGF-beta) and platelet derived growth factors (PDGFs) that aid tumor growth, local invasion, and development of metastases. The hard, fibrotic shell of desmoplasia serves as a barrier to the infiltration of both chemo- and immunotherapy drugs and host immune cells to the tumor. Although there have been recent improvements in chemotherapy and surgical techniques for management of pancreatic cancer, the majority of patients will die from this disease. Therefore, new treatment strategies are clearly needed. CAFs represent an under-explored potential therapeutic target. This paper discusses what we know about the role of CAFs in pancreatic cancer cell growth, invasion, and metastases. Additionally, we present different strategies that are being and could be explored as anti-CAF treatments for pancreatic cancer.

## 1. Background

Progress in the treatment of pancreatic cancer remains slow and tedious. Despite substantial investment in time and resources, the prognosis of patients with pancreatic cancer, primarily pancreatic ductal adenocarcinoma (PDAC), is still poor. In the United States approximately 56,770 patients will be diagnosed with pancreatic cancer each year and 45,750 (80%) will die from it. It is the 4th leading cause of cancer death in both men and women in the world. The most recent American Cancer Society database reports that the 5-year survival of all patients with pancreatic cancer is 9%. If the cancer is diagnosed early enough for the patient to undergo surgery to remove the tumor, the 5-year survival is still only 34% [1]. Therefore, new treatments and new treatment strategies are clearly needed to improve the outcome of patients with this devastating disease.

The pancreatic cancer phenotype is primarily a consequence of oncogenes disturbing the resident pancreas parenchymal cell repair program [2]. The major genes that are mutated in association with pancreatic cancer are KRAS, p16/CDKN2, TP53, and SMAD4/DPC4. BRCA1 and -2 are also associated with an increased risk of pancreatic cancer. The pancreatic cancer milieu consists of transformed cancer cells in various stages of epithelial mesenchymal transition (EMT) and non-transformed stroma. This stroma is composed predominantly of cancer associated fibroblasts (CAFs), as well as macrophages, immune cells, endothelial cells, and epithelial cells [3,4]. The microenvironment plays an important role in influencing tumor progression and prognosis. Each cell type interacts with the transformed cancer cells, and as such is a potential target for treatment. The characteristic *desmoplastic* stroma contributes greatly to the challenges of treating pancreatic cancer; it has been shown in multiple studies to be involved in many aspects of tumor pathogenesis including the promotion of tumor progression, invasion, metastasis, and chemoresistance [5]. EMT is a process of cellular plasticity that contributes to cancer cell invasion and metastasis [6]. CAFs are key players in cancer cell EMT. In a recent study, loss of E-cadherin in tumor buds, increased expression of vimentin, and activation of CAFs, all signs consistent with cancer cell EMT, were associated with more aggressive tumors requiring portal vein resection and an increased probability of positive resection margins [7]. This complexity suggests that an improved understanding of the molecular basis of cell–cell interaction in the cancer stroma is required to effectively target cancer specific growth mechanisms [8,9].

The non-transformed fibroblasts that are both within and surrounding pancreatic cancers are not passive bystanders but rather constitute a complex, active environment (Figure 1) with clear roles in tumor growth and dissemination [10]. CAFs of the pancreatic tumor microenvironment have been shown to enact a dysregulated wound healing response [11] and curiously, have been found to play both tumor-supportive and tumor-suppressive roles [11,12]. The pancreatic CAFs cause fibrosis and desmoplasia that can affect the ability of surgery to excise the tumor and chemotherapy/immunotherapy drugs to eradicate the tumor. Fibrosis induces a firmness and stickiness of the tumor, making it adherent to critical structures and more tedious and challenging to achieve complete tumor excision that is documented pathologically by negative margins. Negative margins at surgery is currently the best chance to cure this disease. Chemotherapy treatment shortcomings have been previously attributed to the desmoplastic stroma. The theory is that CAFs cause desmoplasia that results in decreased microvascularity causing inability of the chemotherapy drugs to effectively penetrate into the tumor [13]. Further complicating this, some studies have shown that the presence of certain sub-types of CAFs are associated with more aggressive tumors and shorter survival suggesting that there is fibroblast heterogeneity in the context of pancreatic cancer and that all CAFs are not the same and do not function in the exact same manner [14].

If we consider all patients who present with pancreatic cancer, nearly two-thirds have distant metastases or locally advanced disease at the time of diagnosis making surgery impossible and upfront chemotherapy critical. However, if we convert locally-advanced pancreatic cancer to completely resectable cancer with preoperative chemotherapy, the long-term survival rate is similar to patients who present with resectable tumors [15]. Chemotherapy unlike radiation therapy does not make the surgery more difficult; therefore, improved chemotherapy regimens are very much needed. Gemcitabine, the long-time drug of choice for pancreatic cancer, only has a response rate of 23% and when given to patients with distant metastases the median survival is only 6 months. An example of anti-fibroblast treatment that has already been shown to improve outcome is (nab)-paclitaxel when it is administered with gemcitabine. When (nab)-paclitaxel is added to gemcitabine, the median survival increases to 9 months. (Nab)-paclitaxel decreases the CAF content in the tumor resulting in a marked alteration of tumor stroma and tissue softening resulting in better penetration of gemcitabine into the tumor cells [16]. Such findings suggest that strategies to affect CAFs may improve outcome of patients with pancreatic cancer and support the use of anti-stromal therapies alongside conventional chemotherapies. The current most effective chemotherapy regimen is FOLFIRINOX which is a combination of leucovorin, 5-fluorouracil, irinotecan, and oxaliplatin. It has a partial response rate of nearly 80% [17,18]. This regimen has been able to convert some locally advanced tumors to surgically resectable [15,18]. However, it is more toxic than other regimens and as such, cannot be used in older patients with poor performance status. Clearly additional improvements in systemic therapy for pancreatic cancer are needed, and the use of anti-stromal strategies may be an effective supporting strategy.

CAFs are the subject of this review and represent an underutilized treatment target in pancreatic cancer. Although some clinical trials and ongoing animal studies are exploring modalities to target CAFs therapeutically [19,20,21,22,23], this area has yet to be fully developed. This paper aims to provide an update on the progress made in understanding pancreas CAFs and their role in stromal formation, cancer progression, invasion, and metastases. We will review mechanisms of stromal influence on pancreatic cancer, chemoresistance, and potential of anti-CAF therapy to improve outcomes.

## 2. CAFs and Tumor Stroma/Extracellular Matrix

Pancreatic cancers are composed of extracellular matrix and various cell types including transformed cancer cells, fibroblasts, endothelial cells, pericytes, macrophages, and a host of other immune cells. Activated CAFs are among the most abundant cell type, comprising between 15–85% of the stromal cells [24,25]. The definition of a cancer associated fibroblast (CAF) is not straight forward [19]. The difficulty comes from a lack of specific cell surface markers that are expressed only on cancer associated fibroblasts and not other cell types. Because of this, CAFs are usually identified by cell morphology, tissue position and lack of lineage markers for other cell types such as epithelial cells, endothelial cells and leukocytes. Expression of vimentin and platelet-derived growth factor receptor-α (PDGFRα) are commonly used as markers for fibroblasts when combined with other criteria like spindle cell shape and tissue location. Other markers for activated fibroblasts also exist, including α-smooth muscle actin (αSMA) and fibroblast activation protein (FAP) [19,26]. Activated fibroblasts may also express podoplanin, fibroblast specific protein-1 (FSP-1), transforming growth factor-beta (TGF-beta), and platelet derived growth factor receptor-beta (PDGFR-beta) [27,28]

CAFs are densely arranged around all cancerous structures as a complete or incomplete ring [5]. Equivalent fibroblasts are not present around benign ducts and other tissue. CAFs present specific activated myoblast-like characteristics such as cellular elongation with pseudopods. TGF-beta (Table 1) has been shown to increase cell stiffness of both normal fibroblasts and CAFs; it also increases CAFs elongation and spreading, which is not seen in standard non-activated fibroblasts. It also enhances CAF invasion in tumor stroma [29].

CAFs are known to generate dense fibrosis or desmoplasia within and around the tumor. A significant proportion of pancreas tumor stroma is composed of various collagen patterns, composing the desmoplasia (Figure 1). Changes in stromal collagen alignment have been shown to modulate cancer cell behavior and have prognostic significance [32]. Highly aligned collagen occurs in 12% of patients with pancreatic cancer and portends an especially poor prognosis [32]. Patients with highly aligned collagen specifically do worse following surgical resection than those whose tumors do not have this characteristic. Tumor associated collagen signature-3 (TACS-3) is characterized by bundles straightened and aligned collagen fibers that are perpendicular to the tumor boundary. TACS-3 is associated with poor disease-free and overall survival in breast cancer [44].

Epigenetic mechanisms (primarily DNA methylation) have been found to regulate genes involved in the formation of extracellular matrix, most commonly hyaluronan. Hyaluronan is an abundant extracellular matrix component in pancreatic cancer and is a marker of poor prognosis. Pancreatic cancer cells can reprogram stromal cells through DNA methylation of genes by direct contact. CAFs actively remodel the desmoplastic stroma by (1) secreting matricellular proteins like hyaluronic acid, collagens, and tenascin C, (2) inducing cross linking of collagen 1 via expression of lysyl oxidase, and (3) secreting metalloproteinases that degrade the matrix. The binding of extracellular matrix proteins (ECM) to integrin receptors on non-tumor cells (immune cells and bone marrow derived dendritic cells) facilitates their recruitment to the ECM and the interaction of the ECM with tumor cells facilitates metastases. CAFs are also known to be highly chemotherapy resistant [45]. High levels of IL-6 correlate with chemoresistance [45,46]. Further, a dense desmoplastic ECM increases interstitial fluid pressure and acts as a barrier to drug delivery, leading to poor accumulation of chemotherapy in tumor resulting in a lack of efficacy.

## 3. PDAC CAF Heterogeneity

CAFs constitute a diverse cell population that is difficult to define as no specific cell surface marker exists, but which is known to consist of several cell subtypes [19] (COX2). In human and mouse pancreatic cancer studies, Tuveson’s group has shown the presence of unique subpopulations of CAFs: Myofibroblastic CAFs (myCAFs) that secrete trophic factors and extracellular matrix components to construct the extracellular matrix and inflammatory CAFs (iCAFs) that secrete Il-6 and use an IL-1 induced signaling cascade that leads to JAK/STAT activation and promotes an inflammatory CAF state [47,48].

CAFs are known to be heterogeneous in pancreatic cancer [35,49,50]. Such heterogeneity exists with regards to cell surface marker expression, cytokine production, cell signaling, cell–cell interactions, and gene expression. This heterogeneity explains why one type of CAF is found to support cancer invasiveness and metastases while another type does not [51]. Single cell technology is being used to delineate CAF heterogeneity in pancreatic cancer, but the exact spatial distribution and forces that drive single-cell phenotypes have not been well described. Using single cell RNA and protein kinetic studies to examine the role of CAFs in modulating heterogeneity of pancreatic cancer, a recent study identifies a shift toward EMT, and fibroblast proliferation linked to mitogen-activated protein kinase (MAPK) and signal transducer and activator of transcription 3 (STAT3) signaling. This shows the impact of stroma in shaping tumor architecture by altering inherent patterns of tumor glands in pancreatic cancer.

Although CAFs are a highly heterogenous cell population they are generally identified as an alpha-SMA positive cell [35,38]. Other markers have been identified in CAFs and expression of each has been associated with an effect on prognosis. There is no one marker; however, that captures all pancreatic cancer CAFs [19]. The heterogeneity of pancreatic cancer CAFs is further supported by recent research that identified a subpopulation of stromal cells that was designated as cancer associated mesenchymal stem cells that caused cancer cell invasion through granulocyte–macrophage colony-stimulating factor (GM-CSF) in one study and increased tumor cellular proliferation and larger tumors in a second study [5,52,53]. GM-CSF appeared to cause these results by inducing down regulation of E-cadherin and up-regulation of TWIST1 and vimentin through the JAK2/STAT3 pathway [5,54]. The relationship between CAFs and these “cancer associated mesenchymal stem cells” has yet to be completely elucidated.

Previously described pathways involved in CAF activation include sonic hedgehog, TGF-beta, IL-1, IL-6, and IL-10 [29,43,55,56,57,58,59]. CAFs are also activated by growth factors and cytokines such as CCL2, PDGF, hepatocyte growth factor, and fibroblast growth factor (FGF) [60]. Once activated CAFs develop a contractile and secretory function, secreting many of the factors known to be further involved in their activation including IL-1 and IL-6 [56]. Some activated CAFs in pancreatic cancer express and secrete fibroblast activated protein (FAP), which influences cancer cell motility and invasion, cancer cell cycle progression, extracellular matrix deposition, and angiogenesis within the tumor matrix [38]. TGF-beta that has been linked to induction of EMT induces FAP expression. The intensity of FAP expression in pancreatic cancer correlates inversely with outcome such that higher expressing tumors have worse outcome [38]. Removing the CAFs expressing fibroblast activation protein (FAP), a protein that inhibits immune cell function, resulted in improved immune control over tumor growth and progression and uncovered the efficacy of immune modulating antibodies [61]. Blockade of FAP in combination with radiation treatment in murine models of pancreatic cancer was associated with antigenic specific tumor T cell infiltrate and enhanced collagen deposition but it did not prolong survival even when combined with anti-PD1 therapy [53]. FAP is also noted to allow pancreatic cancer cells to escape immune surveillance. FAP+ CAFs are the primary pancreatic cancer source of the chemokine ligand 12 (CXCL12). Administration of an inhibitor of chemokine receptor 4, a CXCL12 receptor ligand, resulted in synergizing anti-PD-L1 immunotherapy [31]. Pancreatic cancer immune evasion is well-described and is currently being extensively investigated. It may be overcome by enhancing T cell immune response. It has been shown that despite the presence of antitumor T cells, immunotherapeutic antibodies are ineffective in a murine pancreatic cancer model. Cancer immune suppression in pancreatic cancer appears to be mediated by CXCL 12, the chemokine that binds to cancer cells and excludes T cells by a mechanism that is dependent on the CXCL 12 receptor CXCR4 [31,62] (Table 1).

Some pancreatic CAF sub-populations are now known to have an inflammatory phenotype [28,47]. Podoplanin, a well conserved mucin-type transmembrane protein, exerts various functions including regulation of tissue development and cellular motility. Podoplanin+ fibroblasts are recruited to the pancreas tumor microenvironment. Podoplanin+ fibroblast infiltration of pancreatic carcinoma is associated with worse overall and disease-free survival [39].

## 4. CAFs and Tumor Progression

CAFs and cytokines promote tumor progression. Both play a central role in tumor progression, invasion, and metastases and some fibroblast subtypes may play a role tumor initiation (Table 2). A recent study of the murine intestinal mesenchymal niche demonstrated that colorectal cancer initiation was orchestrated by a population of rare peri-cryptal prostaglandin E2 fibroblasts that exert paracrine control over tumor initiating stem cells [63].

CAFs secrete cytokines and growth factors that stimulate tumor growth. Thymic stromal lymphopoietin (TSLP), a key cytokine for the development of Th2 immunity, is produced by CAFs in pancreatic cancer. Tumor infiltration of Th2 cells is associated with decreased survival. Recent studies show that pancreatic tumor production of IL-1 alpha and beta is a stimulant for TSLP secretion by CAFs [43]. Some CAF-secreted factors are involved in cancer cell EMT. Experimental evidence has suggested a role for zinc finger E-box binding homeobox I (ZEB1) in EMT, invasion, and metastases in pancreatic cancer. ZEB1 expression is present in both cancer cells and fibroblasts. Fibroblast ZEB 1 expression is an independent predictor of survival after pancreatic cancer resection [67]. Asporin is another molecule that is highly expressed in CAFs. It is a mediator of pancreatic stem cell activation and EMT. It mediates invasion and migration of pancreatic cancer cells through both autocrine and paracrine mechanisms [33].

One mechanism by which tumors can recruit and activate fibroblasts is via exosomes [68]. Translocation via exosomes transfers metabolic substrates from CAFs to tumor cells. Exosomes from CAFs contain lactate, acetate, amino acids, lipids, and tricarboxylic acid cycle intermediates that are feeding tumor cells and reprograming them to inhibit mitochondrial oxidative phosphorylation and upregulate glycolysis that allows the cancer cells to grow in an anaerobic environment. Metabolic studies also show that CAF autophagy stimulated by pancreas tumor cells causes alanine secretion that out competes against glutamine for fuel for tumor cells in a low glucose environment. The transcription factor ETV1 doubled pancreatic cancer tumor volume in mice by stromal expansion, altered stromal morphology, increased tumor cell invasion, and upregulated EMT regulators including SLUG, SNAIL, TWIST, vimentin, ZEB1, ZEB2, and MMP9 [69]. A negative regulator of stromal formation, CD 146 or MCAM, has a critical role in tumor progression and invasion. CD 146 negative pancreatic cancer patients have higher grade tumors, more advanced clinical stage and a greater likelihood of postoperative residual cancer. CD 146 positive pancreatic cancer patients have increased survival compared to CD 146 negative patients. Knockout of CD 146 was found to enhance pancreatic cancer cell migration and invasion and induce pro-inflammatory genes SDF1A, CXCL1, CCL5, HGF, and COX2 [70].

## 5. CAFS and PDAC Metastatic Spread

The possible mechanisms by which CAFs can assist in metastases are multiple. CAFs can secrete cytokines and chemokines, as previously mentioned, that specifically support tumor progression. For example, the chemokine CCL5 which acts on cancer cells to promote invasion and metastases [71]. Senescent CAFs, which represent a subtype of CAFs, are known to secrete excess IL-8. IL-8 is a mediator of cancer cell CAF interaction and promotes pancreatic cancer cell invasion and metastasis [36]. Senescent CAFs are also known to inhibit the immune response to the tumor and to remodel extracellular matrix to allow cancer cells to invade. Finally data in human prostate and colon cancer, and a mouse model of lung metastases, suggests that CAFs can circulate in the blood to develop a niche to sustain small numbers of tumor cells that travel through the circulation to distant sites and facilitate tumor cell establishment there [72,73,74]. CAFs also play a pivotal role in inducing metastases by triggering EMT pathways in cancer cells and establishing metastatic niches (Figure 2). Ligorio and others used single cell RNA and protein analytics in a murine model of pancreatic cancer to identify single cell population shifts toward EMT and proliferation linked with MAPK and STAT 3 signaling [35]. Hypoxia marker carbonic anhydrase IX and the lactate transporter MCT 4 in the stroma is associated with EMT transfer and phenotype in pancreatic cancer cells and also portends shorter survival [75]. A recent study showed that portal vein invasion by pancreatic cancer was seen in tumors with loss of membranous E-cadherin in tumor buds, higher expression of vimentin, activated CAF morphology, and margin positive resection [7]. This study suggests that portal vein invasion is associated with aggressive tumor biology and disseminated tumor growth less amenable to margin negative surgical resection.

Malignant cells facilitate lung metastasis by bringing their own stromal components. In an experimental mouse model, the viability of circulating cancer cells was greater when they carried their own heterotopic stromal cell fragments. Moreover, when these circulating tumor cells with stromal components had the cancer associated fibroblasts depleted there was a significant decrease in the number of lung metastases and a prolongation of survival. This demonstrates the importance of CAFs in the development of metastases [36]. Further, stretching fibroblasts remodels fibronectin and enhances the ability of co-cultured cancer cells to migrate. Finally, stretching normal fibroblasts converts them to the phenotype of CAFs suggesting that mechanical stress is a critical factor in fibroblast activation [76]. Pancreatic cancer cell survival and metastases appears to be dependent on crosstalk that is mediated through extracellular vesicles. Formation of an ANXA6/LRP1/TSP1 complex was dependent on CAFs and required physiologic culture conditions that improved tumor cell survival and migration. Depletion of ANXA6 in CAFs results in impaired complex formation and subsequently impaired pancreatic cancer occurrence and metastases. This was reversed by injection of CAF derived ANXA6+ vesicles which enhanced tumorigenesis [77].

CAFs are critical in the metastatic process. CAFs can travel with cancer cells through the blood stream facilitating survival and extravasation at metastatic sites. Aiello et al. showed that fibroblasts appeared in distant metastases when the metastases were as small as 6–7 cells [78]. They also showed that stromal volume of metastases eventually reaches the same volume as the stroma in the primary tumor [79]. In the case of pancreatic cancer liver metastasis, “CAFs” can also be activated locally. Specifically, pancreatic cancer cell exosomes expressing resident surface integrins are taken up by liver Kupffer cells and subsequently activate hepatic stellate cells. Activated hepatic stellate cells acquire a CAF phenotype and contribute to the production of pro-metastatic hepatic niches. Tumor cells then land in the liver and metastases develop. This may explain the early liver metastases seen in some pancreatic cancer patients [80] (Figure 2).

## 6. Modulation of CAFs to Treat Pancreatic Cancer

Most strategies to manipulate CAFs to treat cancer focus on either interfering with the function of CAFs as a barrier to effective drug delivery or inhibiting CAF activation. CAFs have been shown to build cage like structures around pancreatic cancer tumor cells making them resistant to chemotherapy [45]. Transforming growth factor-beta modulates pancreatic cancer cell associated fibroblasts shape, stiffness, and invasion involved in tumor progression and metastases. Inhibition of TGF-beta inhibits these effects and has been shown to decrease virulence of the pancreatic cancer in some studies [29] (Table 3). Inhibition of signaling pathways that are commonly activated in CAFs, like TGF-beta, hedgehog, or the angiotensin II receptor, have been shown to promote a reduction in CAFs, extracellular matrix and a potentiation of drug delivery to tumor. This strategy has inhibited tumor growth and metastases in various pre-clinical studies. A landmark study by Olive et al. showed that depletion of tumor stroma using the hedgehog inhibitor IPI-926 resulted in enhanced gemcitabine-induced intra-tumoral drug accumulation and efficacy in pancreatic cancer [81]. However, subsequent studies including a trial in humans did not show similar positive results [82,83]. Inhibition of CAF-induced pro-tumorigenic signals may be a highly attractive strategy to improve anti-cancer therapy in pancreatic cancer. Researchers have shown that inhibition of the mTOR pathway in pancreas CAFs using the somatostatin analogue SOM230 resulted in reduced CAF mediated secretion of IL-6, a cytokine that stimulates chemoresistance [40]. Importantly, the use of mTOR inhibition potentiated drug sensitivity of the pancreas tumor [40]. It also inhibited CAF-mediated exocrine secretion of IL-6 [40].

Targeting stromal components and stromal depletion has become an area of exciting research in pancreatic cancer (Table 3). Stromally derived lysyl oxidase (LOX) is involved in collagen cross linking and strengthening the collagen matrix. Small molecule LOX inhibitors reduced angiogenesis and growth in experimental breast cancer and led to decreased metastases in lung and liver [90]. In another experimental model of breast cancer, inhibition of LOX activity had no effect on tumor latency and size but significantly decreased tumor metastatis by inhibition of tumor cell extravasation [91]. Thus, inhibition of LOX may be an important potential strategy to inhibit metastasis of cancer. Focal adhesion kinase (FAK) is increased in neoplastic pancreatic cancer cells as well as CAFs [60,92]. It is associated with fibrosis and poor CD8+ T cell infiltration. The selective FAK inhibitor VS-4718 resulted in reduced pancreatic cancer fibrosis and decreased numbers of cytotoxic T cells in a KPC mouse model of pancreas cancer. Treatment with the inhibitor resulted in a doubling of survival and made the mouse model more responsive to T cell immunotherapy and PD-1 antagonists [93]. FAK inhibition also decreased lymph node metastases in a mouse melanoma model by blocking vascular cell adhesion molecule (VCAM-1) [94]. Targeting components of the tumor stroma like fibronectin in pancreatic cancer can effectively reduce tumor growth and spread while enhancing anti-tumor drug delivery [95]. The presence of cancer cellular and stomal fibronectin is necessary for pancreatic cancer progression. Endothelial monocyte activation polypeptide 11 (EMAP II) interferes with fibronectin–integrin angiogenesis signaling in pancreatic cancer. EMAP II caused a significant reduction in tumor growth with a decrease in microvessel density and proliferative activity in a heterotopic pancreatic cancer model. The anti-tumor activity of EMAP II is mediated through targeted interference with stroma fibronectin–integrin dependent pancreatic cancer cell proliferation [96]. New findings indicate that the depletion of fibronectin switches the activity of secreted protein acidic cysteine-rich (SPARC) from promoting cancer cell proliferation to growth inhibition and induction of apoptosis [97].

SPARC is highly expressed in tumor stroma principally in peritumoral fibroblasts. Overexpression of SPARC in this compartment is associated with poorer prognosis. SPARC is involved in angiogenesis, apoptosis, invasion and adhesion, cell cycle progression, and proliferation [86]. SPARC and hyaluronan synthetase are targets of ETV1 that result in stromal expansion. Overexpression of ETV1 increased the incidence and volume of metastases in mouse models. SPARC deletion has been shown to completely abrogate the pro-tumorigenic effects of ETV1 overexpression supporting the pathway link [5,98]. Neuregulin (NRG1) 1, a ligand for HER3 and HER4 receptors, is secreted by both pancreatic cancer cells and CAFs. The desmoplastic stroma adds to pancreatic cancer aggressiveness by promoting tumor progression, invasion, and resistance to chemotherapy. 7E3, an original antibody to NRG1 promotes antibody dependent cellular toxicity in NRG-1+ pancreatic cancer cells and CAFs. It inhibits migration and growth of pancreatic cancer cells that are co-cultured with CAFs both in experimental models of pancreatic cancer. Studies shows that 7E3 could be a promising antibody mediated approach to targeting the relationship between pancreatic cancer and CAFs [99]. Src kinase expression from fibroblasts supports invasion and metastases of pancreatic cancer. Src expression and activity are up-regulated in pancreatic cancer and correlate with reduced survival. The Src kinase inhibitor dasatinib significantly inhibited the development of metastases and growth of pancreatic cancer in a mouse genetically engineered pancreatic cancer model [100]. CAFs have receptors for platelet derived growth factor (PDGF) and are activated by stimulation of their PDGR-receptor. Upon PDGF stimulation and activation of the PDGF-receptor JAK is phosphorylated and further activates STAT3 a transcription factor that is translocated into the nucleus where it promotes the transcription of genes responsible for cell growth, differentiation, proliferation, and apoptosis. The tyrosine kinase receptor antagonist ruxolitinib can inhibit JAK and prevent STAT-3 activation and thus inhibit tumor growth and increase apoptosis of tumor cells [101].

Although the majority of studies suggest that CAFs support tumor progression rather than restraining it, there is still some controversy about whether CAFs are actually good or bad. CAFs form a tumor stromal capsule that may either protect the tumor from adversarial elements like chemotherapy or immunotherapy, but conversely the capsule may inhibit tumor growth. Some studies have shown that depletion of CAFs has induced less differentiated and ultimately more aggressive tumors [20]. Transgenic mice with the ability to delete alpha-SMA myofibroblasts were generated. Depletion of these fibroblasts at either the PanIN or pancreatic cancer stage led to invasive cancerous tumors with EMT and diminished mouse survival. Further, in human pancreatic cancer patients reduced numbers of myofibroblasts in the tumor correlated with reduced survival [20]. Sonic hedgehog (Shh), a soluble ligand overexpressed by pancreatic adenocarcinoma cells, drives formation of fibroblast-rich desmoplastic stroma. Deletion of Shh in a murine model of pancreatic cancer resulted in reduced stromal content of tumor, but unexpectedly, these tumors were more aggressive and had more vascularity, increased proliferation, and undifferentiated histology. Shh deficient tumors could be treated by VEGFR blocking antibody resulting in less tumor angiogenesis and improved survival [22]. Recent studies in mouse models of pancreatic cancer suggest that there may be two functionally different populations of CAFs. One that promotes the tumor and the other that opposes it. A recent study suggests that Meflin, a glycosylphosphatidylinositol-anchored protein that serves as a marker of mesenchymal/stromal cells. In a large number of human pancreatic cancer tissues infiltration by Meflin-positive CAFs correlated with favorable patient outcome. In a mouse model, Meflin deficiency led to tumor progression and poorly differentiated tumors [102]. Expression of Meflin in CAFs may be a method to distinguish a favorable CAF from a detrimental one. This has not yet been elucidated.

In one mouse study, gentle, less complete depletion of CAFs using a biological substance named curcumin that changes CAFs back to a more normal state rather than indiscriminate near complete depletion suppressed CAF-induced pancreatic cancer cell migration and invasion in vitro and lung metastases in vivo [103]. These results indicate possible tumor-suppressive effects of the CAFs in the stroma and indicate that effective anti-CAF therapies in pancreatic cancer will likely need to modify CAFs rather than ablate them entirely. miRNAs, small endogenous non-protein encoding RNAs, are capable of regulating many cellular processes by inhibiting hundreds of genes simultaneously. As such, they may be interesting molecules to modulate the CAF phenotype. Reversal of up regulated or down regulated miRNAs in CAFs using anti-miRNA or mimetic oligonucleotides can result in a non-CAF phenotype. A recent study in pancreatic cancer showed that miR-199a and miR-214 were upregulated in a patient-derived CAFs and that inhibition of these miRNAs caused the dedifferentiation of activated pancreatic stellate cells and inhibited tumor promoting paracrine effects [104]. Another factor to consider is differences in CAF subpopulations, some of which may specifically be tumor-promoting while others are tumor-inhibiting. Current studies must aim to distinguish different sub-types of CAFs such that we can therapeutically affect the detrimental pro-tumoral CAFs types while supporting the anti-tumor ones [5].

CAF effects on tumor immune cells may help to explain their seemingly paradoxical role in pancreas tumors, and these relationships are currently active areas of investigation. The stromal protein Big-h3 is a key factor in the immune paracrine interaction that drives pancreatic cancer. It is produced by CAFs in the stroma of pancreatic cancer. It acts directly on CD8+ T cells and F4/80 macrophages to inhibit the immune anti-tumor response to the pancreatic cancer. Depleting Big-h3 in vivo in mice reduced tumor growth by increasing the number of CD8+ T cells and F4/80 macrophages [37]. Targeting stroma in a murine model of metastatic pancreatic cancer using the immune modulating effect of hyaluronan degradation by PEGPH20 significantly decreased the immunosuppressive effect of CXCL12/CXCR4/CCR7 signaling axis in CAFs, myeloid, and CD8+ T cells. This also resulted in increased CCR7(-) effector memory T cell infiltration, increased interferon-gamma secretion, and increased numbers of CD8+ T cells and most importantly improved survival in an animal model of metastatic pancreatic cancer [105]. Stromal modulation has also shown promising results in the enhancement of immune checkpoint blockade treatment for pancreatic cancer. Inhibiting CXCL12 produced by FAP+ CAFs re-sensitizes pancreatic cancer cells to anti-PDL1 immunotherapy [31]. Effective stromal modulation was recently reported by a polymeric micelle-based nano formulation to deliver a sonic hedgehog inhibitor cyclopamine (CPA) combined with a cytotoxic chemotherapy drug paclitaxel. This combination modulated pancreatic cancer stroma by increasing intra-tumoral vascular density. It then promoted tumor infiltration by cytotoxic CD8+ T cells and when checkpoint blockade with PD-1 was added to this combination prolonged survival in mouse models of pancreatic cancer [89].

## 7. Conclusions

Pancreatic cancer CAFs represent an exciting area of research and many excellent preliminary studies suggest that this will be an important area for anti-tumor therapy. Our understanding of the heterogeneity, activation, and roles of pancreas CAFs needs to improve in order to maximize their potential for therapeutic benefit. Because there is convincing evidence for the heterogeneity of CAFs in pancreas tumors and sub-populations with differing functions, it will be necessary to target specific CAF subsets to achieve clinically relevant anti-cancer effects. This means that we will need to better identify specific CAF targets. It may also be possible to therapeutically target CAFs in metastases, since they are known to play an important role in this context. The future is to target specific subgroups to produce the anti-tumor effects needed either in concert with chemotherapy or the immune system. This may be an effective strategy to increase the cure-rate in patients with pancreatic cancer.

## Figures and Tables

**Figure 1 cancers-12-01347-f001:**
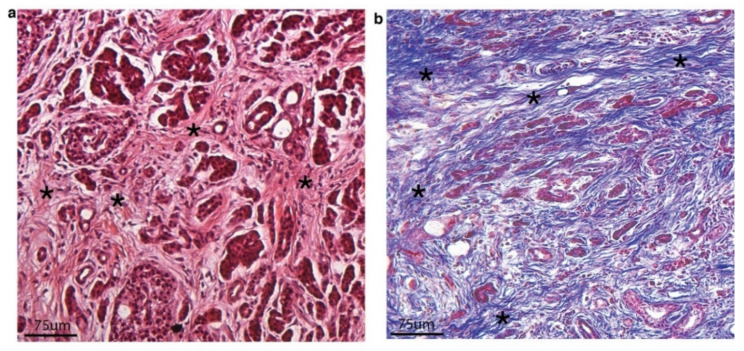
(**a**) Hematoxylin and Eosin (H&E stain) of a surgically resected pancreatic ductal adenocarcinoma showing small and medium glands with irregular morphology embedded in dense, desmoplastic stroma (highlighted with black asterisks). (**b**) Trichrome stain of surgically resected pancreatic ductal adenocarcinoma highlighting severe desmoplasia and dense matrix that appears as linearized ribbons of blue stain (collagen fibers) (highlighted with black asterisks).

**Figure 2 cancers-12-01347-f002:**
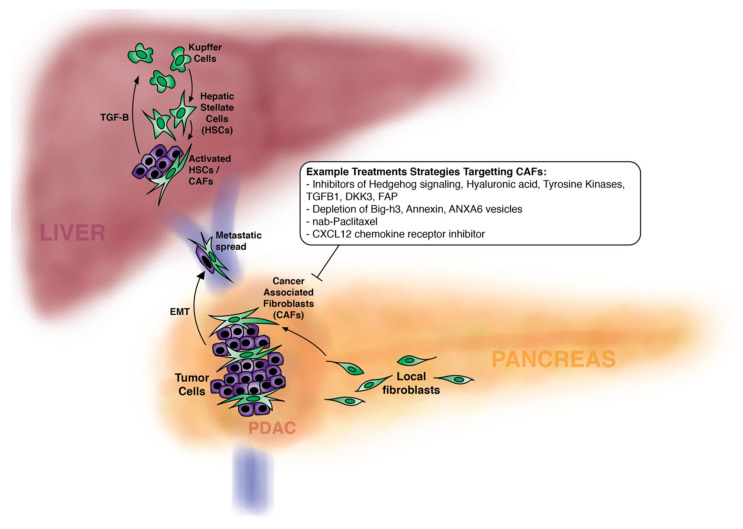
Primary pancreatic cancer recruits fibroblasts to become cancer associated fibroblasts (CAFs). Tumor cells with CAFs invade the portal vein and metastasize to the liver. In order to establish a focus in the liver, Kupffer cells secrete transforming growth factor-beta (TGF-beta) that recruits and activates hepatic stellate cells to become CAFs. Strategies to treat pancreatic cancer through CAFs include inhibitors of the hedgehog pathway, hyaluronidase, tyrosine kinase, and inhibitors of CAF recruitment. miRNAs are a potentially useful strategy because such tools can inhibit multiple pathways, simultaneously. Nab-Paclitaxel has been shown to target pancreatic cancer CAFs. AMD3100, a CXCL12 chemokine receptor inhibitor, shows improved immune response/immunotherapy response to pancreatic cancer—CXCL12 is secreted by CAFs.

**Table 1 cancers-12-01347-t001:** Cancer associated fibroblasts (CAFs) play a central role in the multistep processes of tumor initiation, progression, invasion, and metastases.

Molecule	Source Cell	Target Cell	Effect	Reference
Annexin A6	CAFs	Cancer cells	CAF derived annexin A6+ extracellular vesicles found to support pancreatic cancer aggressiveness	Leca et al., J Clin Invest, 2016 [30]
CXCL12	CAFs	Cancer cells	Effector of immunosuppression by FAP+ CAFs	Feig et al., Proc Natl Acad Sci, 2013 [31]
Collagen	CAFs	Cancer cells	Well aligned collagen in the stroma is associated with worse prognosis	Drifka et al., Oncotarget, 2016 [32]
Asporin	Activated pancreatic stellate cells, CAFs	Cancer cells	Enhances EMT, promotes cancer cell invasion and metastases	Wang et al., Cancer Lett, 2017 [33]
TGF-B	CAFs, Cancer cells	CAFs, Cancer cells, CD8 T-cells	Increases CAF stiffness and elongation Suppresses CD8 T cell acquisition to the tumor as well as function	Stylianou et al., Biochim Biophys Acta Gen Subj, 2018 [29] Ahmadzadeh & Rosenberg, J Immunol, 2005 [34]
MAPK, STAT3 signaling	CAFs	Cancer cells	Paracrine CAF TGF-B promotes MAPK and STAT3 signaling, which causes EMT and enhanced PDAC proliferation	Ligorio et al., Cell, 2019 [35]
Il-8	Senescent CAFs	Cancer cells	Prometastatic phenotype	Wang et al., 2017. [36]
Big-h3	CAFs	Tumor CD8+ T-cells	Inhibits tumor specific CD8 T-cells and increases tumor growth	Goehrig et al., Gut, 2019 [37]
FAP	CAFs	Cancer cells	Cancer cell motility, invasiveness and progression. Also, tumor angiogenesis and ECM deposition.	Kawase et al., BCM Gastroenterol, 2015 [38]
Podoplanin	CAFs	Cancer Cells	Worse outcome	Hu et al., Cell Physiol Biochem, 2018 [39]
Il-6	CAFs	Cancer cells	Survivin (apoptosis inhibitor) expression	Duluc et al., EMBO Mol Med, 2015 [40]
VEGF	Pancreatic stellate cells	Tumor stroma	Angiogenesis	Masamune et al., Am J Physiol Gastrointest Liver Physiol, 2008 [41]
SDF-1	CAFs	Cancer cells	Tumor progression and resistance to gemcitabine	Wei et al., Cell Death Dis, 2018 [42]
Thymic stromal lymphopoietin (TSLP)	CAFs, simulated by cancer cell Il-1	Immune cells	Development of Th2 immunity, worse survival	Brunetto et al., J Immunother Cancer, 2019 [43]

**Table 2 cancers-12-01347-t002:** Primarily pancreatic ductal adenocarcinoma (PDAC) CAF subtypes and their characteristic markers and functions.

PDAC CAF Sub-Type	Characteristics	Proposed Role	Reference
iCAF (inflammatory CAF)	Il-6^high^, aSMA^high^, adjacent to tumor cells	Tumorigenesis and cancer progression	Ohund et al., JEM, 2017 [48]
myCAF (myofibroblasts CAF)	Il-6^low^, aSMA^low^, distant from tumor cells		Ohund et al., JEM, 2017 [48]
apCAFs (antigen presenting CAFs)	Express CD74 and MHC class II	Activate CD4 T-cells	Elyada et al., Cancer Discov, 2019 [64]
FAP+ CAFs	Express FAP	Escaping the immune system (blocking of CD8+ anti-tumor T cells)	Zhang et al., Oncotarget, 2016 [65]
CD10+GPR77+ CAFs		Enhances PDAC cell invasion	Su et al., Cell, 2018 [66]
Podoplanin+ CAFs	Express podoplanin	Worse prognosis	Hu et al., Cell Physiol Biochem, 2018. [39]
aSMA+ PDAC myofibroblasts	Express aSMA	Depletion of this sub-type resulted in more aggressive tumors and decreased survival in mice	Ozdemir et al., Cancer cell, 2015 [20]
PDGFRa+ SAA1+ CAFs	Express PDGFRa and SAA1	Stimulate PDAC tumor growth in mice	Djurec et al., Proc Natl Acad Sci, 2018
“Sub-type A”	aSMA^low^, Vimentin^low^, Proliferation^high^, ECM+	“invasive front”, poor prognosis	Neuzillet et al., J Pathol, 2018 [50]
“Sub-type B”	aSMA^high^, Vimentin^high^, Proliferation^low^, ECM+	Intermediate prognosis	Neuzillet et al., J Pathol, 2018 [50]
“Sub-type C”	ECM+, Immune++	Good prognosis	Neuzillet et al., J Pathol, 2018 [50]
“Sub-type D”	aSMA^high^, Vimentin^high^, Proliferation^low^, ECM+	Poor prognosis	Neuzillet et al., J Pathol, 2018 [50]

**Table 3 cancers-12-01347-t003:** Anti-CAF treatment strategies and results in experimental models.

Treatment	Result	Reference
Depletion of CAF-derived annexin (67)	Impaired tumor cell survival and migration in mouse model	Leca et al., J Clin Invest, 2016 [30]
Sonic hedgehog inhibitor (cyclopamine, CPA) PD1 checkpoint blockade Sonic hedgehog inhibitor (vismodegib)	Tumor infiltration by cytotoxic CD8+ T cells and prolongation of survival in murine models Decreases desmoplasia, improves effects of chemo- and cancer nano-therapies in mice	Zhou et al., Biomaterials, 2018. [55] Mpekris et al., J Control Release, 2017 [84] Hedgehog inhibitor clinical trials are ongoing
Depletion of Big-h3	Reduction of pancreatic tumor growth by functionally reprogramming F4/80 macrophages in tumor environment	Goehrig et al., Gut, 2019. [37]
Treatment with Nutilin-3a induces p53 activation	Induces p53 activation in the stroma, reverses activation of pancreatic stellate cells, and decreases stromal fibrosis	Saison-Ridinger et al.m PLoS One, 2017 [85]
Anti-SPARC with Nab-paclitaxel	Positive results in clinical trials vs. human pancreas cancer	Multiple clinical reviewed in the following: Vaz et al., Pancreas, 2015 [86]
miRNA therapies that target ZEB and its downstream pathway	Strategy to improve outcome by inhibiting multiple gene pathways	Bronsert et al., Surgery, 2014 [67]
PEGPH20 (digests hyaluronic acid)	Improves delivery of immuno- and chemotherapy to the tumor	Wong et al., Curr Oncol Rep, 2017 [87] Clinical trials ongoing
Galunisertib (TGFB-1 inhibition)	Inhibits TGFB-1 receptor, found to show improved overall survival in combination with gemcitabine versus gemcitabine alone.	Melisi, Br J Cancer, 2018 [88] Clinical trials ongoing
Depletion of ANXA6 extracellular vesicles in CAFs	Impaired pancreas cancer migration and invasion.	Leca et al., J Clin Invest, 2016 [30]
AMD3100 (a CXCL12 chemokine receptor inhibitor)	Resulted in rapid T cell accumulation in tumor that synergized with PDL1 to destroy pancreas cancer cells	Feig et al., PNAS, 2013 [31]
DKK3 blocking monoclonal antibody	Inhibited pancreas cancer progression, tumor growth, and prolongs survival in mouse model	Zhou et al., ACS Nano, 2018 [89]

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
