# Peer review of "Pancreatic Cancer Associated Fibroblasts (CAF): Under-Explored Target for Pancreatic Cancer Treatment"

_cancers, 2020, doi:10.3390/cancers12051347_

Round 1

Reviewer 1 Report

This manuscript by Norton et al. has reviewed a number of papers on the role of fibroblasts in the progression of pancreatic cancer, which is one of the hottest topics in the field. The manuscript has covered many aspects of cancer-associated fibroblasts (CAFs), particularly focusing on their cancer-promoting roles. I believe that the manuscript is well written so that non-English native researchers can understand it. I found that it was useful to me to overview the field, and believe it helps readers in various fields to understand the significance of tumor stroma in pancreatic cancer development, progression and treatment. I agree with the authors’ important description in the Conclusion section that “Because there is convincing evidence for the heterogeneity of CAFs in pancreas tumors and sub-populations with differing functions, it will be necessary to target specific CAF subsets to achieve clinically relevant anti-cancer effects”. I agree that this manuscript is a very sound paper and merits publication in the journal and do not have any major concerns except one thing described below.

As the authors stated, recent evidence reported by several famous laboratories has suggested that some fibroblasts promote PDAC progression whereas others suppress its progression. Although the authors cited the milestone paper published from Raghu Kalluri’s group (Özdemir et al., ref 61) that first showed the evidence of the existence of fibroblasts that retard cancer progression, they did not cite another milestone paper published in the same issue of Cancer Cell as Özdemir et al., which is the paper by Ben Stanger’s lab (Andy Rhim et al., Cancer Cell 25:735-747, 2014). This paper is also the first one that demonstrated the possibility that there exist fibroblasts that suppress the progression of PDAC in a mouse model. I recommend that the authors cite the paper by Rhim et al. and put it in the description of CAF heterogeneity. Another interesting recent paper in the filed is the paper by Mizutani et al. (Cancer Res 79:5367-5381, 2019), which more extensively characterized fibroblasts that have a tumor-suppressive role in mouse PDAC models and human patients. They first identified “meflin” as a functional marker of cancer-restraining CAFs. I think this paper should also be described in the manuscript, give the authors’ and readers’ interest in the heterogeneity of CAF function.

Minor issue:

Page 3, bottom

myoblastic-like should be myofibroblast-like?

Author Response

Thank you so much for your careful review of our manuscript # 775942 entitled "Pancreatic cancer Associated Fibroblasts (CAF): under-explored target for pancreatic cancer treatment". We have carefully reviewed the manuscript to deal with each of the criticisms of the two reviewers. I will now list a point by point response to the reviewers suggestions.

In response to reviewer # 1, As the reviewer stated, we have clarified the observation that some fibroblasts promote PDAC progression whereas others suppress it. we have included the milestone paper published from Raghu Kalluri’s group (Özdemir et al., new reference 20) that first showed the evidence of the existence of fibroblasts that retard cancer progression. As suggested, we cited another milestone paper published in the same issue of Cancer Cell by Ben Stanger’s lab (Andy Rhim et al., Cancer Cell 25:735-747, 2014, new reference 22). This paper demonstrated that there exist fibroblasts that suppress the progression of PDAC in a mouse model. We cited the paper by Rhim et al. and put it in the description of CAF heterogeneity. We also cited a recent paper by Mizutani et al. (Cancer Res 79:5367-5381, 2019, new reference 100) , which more extensively characterized fibroblasts that have a tumor-suppressive role in mouse PDAC models and human patients. This paper identified “Meflin” as a functional marker of cancer-restraining CAFs. We want to thank the reviewer for guiding us to these very important papers.

On page 3 at the bottom, we changed myoblastic-like to myofibroblastic-like as suggested.

Reviewer 2 Report

Norton et al, have compiled in this review the latest works that focus on a very important aspect involved in pancreatic cancer progression. They have synthesized meaningful information on how tumor stroma generated by CAFs modulates different steps of the metastatic cascade. Taking into account the increasing number of anti-stromal therapies that are being developed in the recent years, this reviewer finds the topic of this review timely and appropriate. Nonetheless, several issues should be fixed before its publication.

  1. The authors use the term “pancreas cancer” throughout the entire manuscript. I suggest the usage of “pancreatic cancer” instead.

  1. The classic abbreviation rules should be revised.  For example PDAC was never defined.

  1. Background, paragraph 2: Mentioning the main important gene mutations driving pancreatic cancer can be of interest.

  1. The definition of the identity of a “CAF” is rather controversial in the field. Although no specific marker strictly exists, authors can refer to a recent consensus: Sahai et al, Nat Rev Cancer, 2020.

  1. Table 1 needs reorganization. The cell type producing certain molecules and its target cell types are sometimes rather confusing. For instance, it is not specified which is the cell type in charge of producing MAPK/STAT3 or IL8, Big-H3, podoplanin… Adding an extra column to specify the producer cell type can help for a better understanding of this table. Same for target cell type.

  1. Collagen alignment to the perimeter of cancer cell clusters or buds is a well established readout of tumor invasiveness, and correlates with the progression status of the tumor, with curly randomly organized collagen fibers representing a rather benign state, and with more perpendicularly aligned fibers being a bad prognostic factor. Even if authors make a good point on this in part 2 of the review, I think it is worth mentioning Dr Keely’s Tumor Associated Collagen Signatures (or TACS), initially described in breast cancer.

  1. In part 3 the authors talk about CAFs heterogeneity in PDAC. Although many of the different subtypes that have been identified are discussed, I think that more emphasis could be made on some of the main PDAC CAFs sub-classifications. Myofibroblastic (myCAF), inflammatory (iCAF) and antigen presenting (apCAFs), described by Dr Tuveson’s group could be discussed in this part of the review.

  1. The review could benefit from a table including the different studies of PDAC CAF subtypes and their characteristic markers/functions.

  1. Authors mention that FAP blockade in combination with radiation therapy improves survival. However, the efficacy is very limited, if any, in the study that they mention. These conclusions should be reformulated.

  1. In part 4, the authors discuss about the role of CAFs as tumor drivers. However, referring to “tumorigenic drivers” can be misleading. Using this term may lead to wrong interpretation, as some readers will associate this with the induction of driver mutations leading to tumorigenesis, which is definitely an event that occurs before the appearance of CAFs and its derived desmoplasia. The authors should consider renaming this section of the review to talk about tumor progression in general. As well, the authors claim that CAFs “play a central role in the multistep process of tumor initiation…”. To my knowledge, the only reference of an example where fibroblasts (not CAFs) can help initiating tumorigenesis is a very recent paper from Roulis et al, Nature, 2020 (in colon cancer). Other specific references should be included for this, if they exist.

  1. In part 5, authors discuss about CAFs and metastasis formation. They mention a study related to PVI. In this study low E-cadherin levels in tumor buds is associated with PIV, so what the authors described in the review is not correct. As well, although if both E-cadherin and Vimentin are markers of EMT, E-cadherin is a negative regulator. This whole concept should be stated clearly in the review.

  1. Figure 2: The authors might consider revising the resolution of the images. As well, even if I think that the figure is informative, the authors might consider avoiding vague statements like miRNA or inhibition of CAF recruitment, and including some specific examples of each.

  1. One very interesting point that the authors comment is the fact that CAFs can travel with CTCs facilitating extravasation and metastasis. Although appealing, this point should be addressed carefully, as, to my knowledge, only 2 studies (Ao et al, Cancer Research, 2015; Jones et al, Prostate, 2013) have reported the existence of CAFs in the bloodstream, in colon and prostate human cancer, respectively. One study showed the presence of circulating CAFs also in a mouse model of lung metastasis (Duda et al, PNAS, 2010). In the study that the authors cite (Ligorio et al, Cell, 2019), although very interesting, there’s no such claim. References should be updated or alternatively, the text should be rephrased.

  1. The authors discuss in part 6 several strategies that have been described to modulate the stromal CAF compartment in order to improve PDAC treatment. This section and table 2 provide in depth information about therapeutic options. However, as the main role of CAFs is the production and rearrangement of ECM, I think it is worth considering the inclusion of other ECM-linked therapies such as LOX, FAK and Fibronectin inhibition, among others.

  1. Although the majority of reports in the field suggest that CAFs promote tumor progression rather than restraining it, there is still some controversy. As authors mentioned, CAFs form a stromal capsule that can physically constrain tumor growth to an extent. However, therapies that aim to eliminate CAFs would probably eliminate this protective effect. The authors could elaborate more on this duality.

  1. The authors discuss about hepatic stellate cells during pre-metastatic niche formation, without citation. Probably they are referring to Costa-Silva et al, NCB 2015. This should be cited.

  1. As the main focus of this review is related to the therapeutic opportunities based on CAF targeting, I think that a table including past and running clinical trials could be of interest for the readers. Some of them are already mentioned in the last part within the text, however an extended table can provide more comprehensive and structured information.

One of my main concerns with this review, which appears constantly throughout the entire manuscript, is related to citations. The authors should check thoroughly all their references to provide clear information to the reader. Specific examples (amongst many others) are:

  • Some references appear twice in the ref list (refs 19 and 39 are the same, for instance).
  • There are many spelling mistakes in the reference list.
  • Many references are misplaced (For example, section 2, citation 22 does not correspond with the claim of the authors regarding complete or incomplete rings). All the others should be revised.
  • The authors make claims that are often not cited. This should be revised.
  • The authors should try to cite original research rather than reviews when talking about specific experimental findings.

Author Response

In response to reviewer #2, first of all we want to thank her or him for the very diligent review of the manuscript and the numerous suggestions that each of which greatly strengthens the work.

(1) As suggested, we have changed pancreas cancer to pancreatic cancer throughout the entire manuscript.

(2) We have tried to carefully define each abbreviation when it is first used throughout the entire manuscript.

(3) Background paragraph 2, we have listed the gene mutations driving pancreatic cancer as suggested

(4) As the reviewer suggests the definition of the identity of "CAF" is controversial.  However, we referred to a recent consensus conference (Sahai et al, Nat Rev Cancer, 2020, new reference 19, section 2) to better illuminate the definition of CAF

(5) As suggested, we have completely revised Table 1 to clarify the cell type that produces the molecule that is identified and the target cell for each molecule.

(6) As suggested, collagen organization and alignment is important and we have referenced Dr Keely's Tumor Associated Collagen Signatures (TACS) (new reference 44, section 2) that are important for prognosis in breast cancer.

(7) As suggested, CAF heterogeneity in PDAC is important and many different subtypes have been described.  However, as suggested we emphasized Dr Tuveson's group work including myofibroblastic (myCAF), inflammatory (iCAF), and antigen presenting (apCAF) as different CAF subtypes with specific different functions (section 3, new references 47 and 48 and Table 2).

(8) As suggested, we included a new table with different CAF subtypes and characteristic markers and function (Table 2).

(9) As suggested, we reformulated the remarks and conclusions about the efficacy of FAP blockade and radiation therapy because of the limited efficacy.  It did not prolong survival (reference 53, section 3).

(10) As suggested, we eliminated the words "tumorigenic drivers" because it can lead to the wrong interpretation and be misleading as tumorigenesis occurs before the existence of CAFs.  Further we renamed this section tumor progression as suggested (section 4).  We did reference the recent paper by Roulis et al in Nature 2020 (reference 66) about the role of fibroblasts and not CAFs help in tumorigenesis in colon cancer.

(11) As suggested, we further clarified the role of CAFs in portal vein invasion and development of liver metastases.  Low E-cadherin levels in tumor buds is associated with portal vein invasion.  Further, if both E-cadherin and vimentin are markers of EMT, we made it clear that E-cadherin is a negative regulator as the reviewer indicates (reference 7, section 5 just prior to Figure 2).

(12) As suggested, we improved the resolution of Figure 2 and we deleted the statements about MiRNA and inhibition of CAF recruitment.

(13) The interesting point that CAFs can facilitate extravasation and metastasis of tumor cells is addressed more carefully as suggested.  As suggested, we removed the citation of Ligorio et al, because it did not cover this topic.  We included the two suggested references that reported the presence of both CAFs and cancer cells in the blood stream in both colon and prostate cancer respectively (Ao et al, Cancer Research 2015 new reference 73, Jones ML et al, new reference 72). As suggested, we also added the presence of circulating CAFs in a mouse model of lung metastasis (Duda et al, PNAS, 2010, new reference 74).

(14) As suggested, we added inclusion of ECM-linked therapies as possible treatment options to Table 3 and part 6 on treatment strategies.  We included some discussion of LOX, FAK and fibronectin inhibition as methods to inhibit the production and re-arrangement of ECM and its potential effect on tumor.

(15) As suggested, we elaborated on the controversy about the role of CAFs as either promoting tumor progression or restraining it.  We mentioned that CAFs form a stromal capsule that may either protect or constrain tumor.  However, therapies to eliminate CAFs may eliminate the protective effect.

(16) As suggested, we added the reference (Costa-Silva et al, NCB 2015 reference 80) about hepatic stellate cells during pre-metastatic niche formation for liver metastasis.

(17) As suggested, we greatly amplified Table 3 to try to better elucidate the trials associated with anti-CAF to improve outcomes pancreatic cancer

We have tried to revise the references as suggested making them accurate, eliminating multiple citations of the same reference and focusing on original research rather than review papers.  We have also tried to use the correct reference to support our claims.  We have tried to eliminate all the spelling mistakes in the references and the manuscript.

We want to thank both reviewers for their very careful review and input about the manuscript.  We believe that we have considered and tried to effectively revise the manuscript in accord with each of the reviewers remarks and criticisms.  We hope that the manuscript is now acceptable for publication in Cancers.

Sincerely,

Jeffrey Norton

Stanford University School of Medicine

Round 2

Reviewer 2 Report

The authors have successfully addressed all my concerns. I believe that the review is now suitable for publication in Cancers.